# The Re-Identification of Previously Unidentifiable Clinical Non-Tuberculous Mycobacterial Isolates Shows Great Species Diversity and the Presence of Other Acid-Fast Genera

**DOI:** 10.3390/pathogens11101159

**Published:** 2022-10-07

**Authors:** Yanua Ledesma, Gustavo Echeverría, Franklin E. Claro-Almea, Douglas Silva, Salomé Guerrero-Freire, Yeimy Rojas, Carlos Bastidas-Caldes, Juan Carlos Navarro, Jacobus H. de Waard

**Affiliations:** 1Laboratorios de Investigación, Facultad de Ciencias de Salud, Universidad de Las Américas (UDLA), Quito 170125, Ecuador; 2Instituto de Investigación en Salud Pública y Zoonosis-CIZ, Universidad Central del Ecuador, Quito 170521, Ecuador; 3Programa de Doctorado, Facultad de Ciencias Veterinarias, Universidad de Buenos Aires, Buenos Aires C1428EGA, Argentina; 4Servicio Autónomo Instituto de Biomedicina Dr. Jacinto Convit, Universidad Central de Venezuela, Caracas 1010, Venezuela; 5Grupo de Microbiología Aplicada, Universidad Regional Amazónica Ikiam, Tena 150102, Ecuador; 6One Health Research Group, Facultad de Ingeniería y Ciencias Aplicadas, Biotecnología, Universidad de las Américas, Quito 170125, Ecuador; 7Programa de Doctorado en Salud Pública y Animal, Facultad de Veterinaria, Universidad de Extremadura, Cáceres 10003, España; 8Grupo de Enfermedades Emergentes, Ecoepidemiologia y Biodiversidad, Facultad de Ciencias de la Salud, Universidad Internacional SEK, Quito 170107, Ecuador

**Keywords:** Ziehl-Neelsen (ZN) staining, partially acid-fast bacilli (AFB), non-tuberculosis mycobacteria (NTM), Nocardia puris, Tsukamurella pulmosis, Gordonia sputi

## Abstract

Non-tuberculous mycobacteria that cannot be identified at the species level represent a challenge for clinical laboratories, as proper species assignment is key to implementing successful treatments or epidemiological studies. We re-identified forty-eight isolates of Ziehl–Neelsen (ZN)-staining-positive “acid-fast bacilli” (AFB), which were isolated in a clinical laboratory and previously identified as *Mycobacterium* species but were unidentifiable at the species level with the hsp65 PCR restriction fragment length polymorphism analysis (PRA). As most isolates also could not be identified confidently via *16S*, *hsp65*, or *rpoB* DNA sequencing and a nBLAST search analysis, we employed a phylogenetic method for their identification using the sequences of the *16S* rDNA, which resulted in the identification of most AFB and a *Mycobacterium* species diversity not found before in our laboratory. Most were rare species with only a few clinical reports. Moreover, although selected with the ZN staining as AFB, not all isolates belonged to the genus *Mycobacterium*, and we report for the first time in Latin America the isolation of *Nocardia puris*, *Tsukamurella pulmosis*, and *Gordonia sputi* from sputum samples of symptomatic patients. We conclude that ZN staining does not differentiate between the genus *Mycobacterium* and other genera of AFB. Moreover, there is a need for a simple and more accurate tree-based identification method for mycobacterial species. For this purpose, and in development in our lab, is a web-based identification system using a phylogenetic analysis (including all AFB genera) based on *16S* rDNA sequences (and in the future multigene datasets) and the closest relatives.

## 1. Introduction

Non-tuberculous mycobacteria (NTM) are facultative intracellular pathogens that in general are not transmitted from person to person but acquired by susceptible individuals from the environment [1,2]. Recently, however, it has also been shown that NTM infections can be acquired through transmission, via fomites and aerosols [3]. Approximately 200 NTM species have been identified, of which approximately 140 are pathogenic to humans and animals [4]. 

Accurately identifying NTM at the species level is essential for providing a correct disease diagnosis, identifying expected antimicrobial susceptibility patterns, and tracing outbreaks for infection source analyses [5,6,7,8,9]. The Ziehl–Neelsen (ZN) staining method is the first step in identifying mycobacteria, since they show up as “acid-fast bacilli” (AFB), a physical property that gives this bacterium the ability to resist decolorization by acids during staining procedures. Even though acid fastness can be attributed to many different genera of bacteria (the genus *Nocardia*, for example), in clinical practice it is a fairly unique characteristic of *Mycobacterium* species. The further identification of AFB at the species level has become complicated because of the steady increase in species in the last 10 years. The phenotypic approaches based on morphological characteristics, growth rates, preferred growth medium, pigmentation, and biochemical tests are time-consuming and nowadays useless [10,11]. Several commercial molecular identification methods are available but these are relatively expensive and are only used in laboratories with enough resources [12]. Small regional laboratories have chosen alternative methods for identification, and a popular, fast, cheap technique—described for the first time about 30 years ago in 1993—is the PCR restriction fragment length polymorphism analysis (PRA) method, targeting a fragment of the heat shock protein 65 (*hsp65*) gene [13]. This method has been shown to be adequate for identifying the most important species in a convenient way [14]. For less prevalent mycobacterial species, the technique fails because of the presence of identical or near-identical PRA patterns of other mycobacteria species in the database [15,16]. 

The present study aimed to identify 48 unidentified NTM species isolated in a tuberculosis diagnostic laboratory in Caracas, Venezuela. Re-identification was considered important, as these strains were of clinical relevance. Forty-five strains were isolated from patients with respiratory problems or other tuberculosis-like clinical manifestations. Three strains were isolated from environmental samples and were associated with NTM outbreaks [9]. The laboratory had tried to identify these AFB with the PRA technique, but this technique either failed to assign a definitive species name, the restriction enzyme pattern could not be found in the PRA database [16], or no *hsp65* PCR signal was obtained and the isolates were reported as *Mycobacterium* spp. to the physician. Here, we sequenced the *16S rRNA*, *rpoB*, and *hsp65* genes of these strains and used a phylogenetic analysis for their identification. 

## 2. Materials and Methods

### 2.1. Acid-Fast Mycobacterial Strains

The 48 strains came from a strain bank present in the ‘Dr. José María Vargas’ Hospital in Caracas and were isolated from 2014 to 2021 from patients with a presumed tuberculosis infection (42 strains were isolated from sputum, pus, biopsies, pleural liquid, or urine samples). Three strains were isolated from a water sample, food, or cosmetic product; all products were associated with NTM outbreaks [8,9]. Three strains were isolated from patients, but the type of clinical sample was unknown as these strains were referred to our lab for identification. The strains were isolated on Lowenstein–Jensen (L-J) medium and all were ZN-positive; for that reason, they were classified as mycobacteria. Most of these 48 isolates were rapid growers with the exception of 6 strains that were slow growers. The previous identification was unsuccessfully carried out with the PRA technique, obtaining an uncertain or ambiguous result after digestion with restriction enzymes. Three inoculation loops of growth of the strains on L-J medium were stored in TE buffer (for DNA isolation) and in TSB with 10% glycerol at −20 °C or −70 °C, respectively. See Appendix A for more detailed information regarding the origin of the strains, type of clinical sample from which the strains were isolated, year of isolation, and preliminary identification (where available).

### 2.2. PCR and Sequencing

The DNA was liberated from the 48 strains by boiling the strains in TE buffer for 5 min. After a brief centrifugation step (1 min at 10,000× *g*), the supernatant was used for the PCR reaction. The PCR amplification of three genes (*16S*, *hsp65*, and *rpoB*) was performed with GoTaq® Green Master Mix from Promega®. The primers for the amplification of the genes can be found in Table 1. The PCR conditions have been described previously [17]. The amplicons were visualized via electrophoresis in a 1% agarose gel, and when specific products were amplified, these products were sequenced with the Sanger method and analyzed with the ABI 3500xL Genetic Analyzer from Applied Biosystems at the Service Department of Universidad de las Americas, Quito, Ecuador.

### 2.3. Identification with a Phylogenetic Analysis 

The sequences of the *16S rRNA*, *rpoB,* and *hsp65* genes, obtained with forward and reverse primers, were assembled and consensus sequences were edited using MEGA 11. The identity of each sequence was confirmed using BLAST in NCBI resources. For the phylogenetic analysis, a total of 40 sequences deposited in GenBank NCBI from 40 species of *Mycobacterium* were included as related groups. *Tsukamurella tyrosinosolvens, Tsukamurella pulmonis, Nocardia puris, Nocardia cyriacigeorgica, Gordonia sputi, Gordonia iterans,* and *Corynebacterium jeikeium* were selected as outgroups (GenBank Accession numbers can be found in Appendix A). The DNA sequence matrices of 88 sequences x 497 nucleotides for *16S*, 80 sequences x 288 nucleotides for *rpoB*, and 69 sequences x 389 nucleotides for *hsp65* were aligned using MEGA 11 [20,21] with the ClustalW algorithm with high gap creation and extension penalties of 16.0 and 6.0, respectively, searching for a strong positional homology. 

Afterward, a maximum parsimony (MP) analysis was implemented using the heuristic search option with a tree bisection reconnection branch-swapping algorithm with a random stepwise addition of 10 replicates for each search and 100–1000 replications per analysis. The gaps were treated as missing data. The characters were treated as unordered and equally weighted. We also performed a maximum likelihood (ML) and substitution models estimated on MEGA 11 (Appendix A) using an MP as a guide tree. In both analyses, the robustness of the trees was estimated using parsimony bootstrapping with 500 pseudo-replicates. For classification purposes for complexes and groups, and for the verification of phylogenetic species, each monophyletic clade was evaluated by means of the divergence between and within clades (uncorrected p-distance calculated), as well as between sequences identified in the NCBI and what was obtained in the laboratory. 

### 2.4. Ethical Considerations

The Institutional Review Board of Servicio Autónomo Instituto de Biomedicina Dr. Jacinto Convit in Caracas, Venezuela, approved the study protocol with a waiver of informed consent (IRB SAIB 06-06-2022).

## 3. Results

### 3.1. PCR and Sequencing

All 48 strains yielded a sequence of the *16S rRNA* and *rpoB* genes and only 43 strains yielded an *hsp65* sequence, as five strains did not amplify with the *hsp65* primers TB11 and TB12 (LTM4391, LTM626213, LTN3490, LTL3092, and LTQ5825). The sequences were introduced in GenBank for preliminary identification. The nucleotide sequence data of these strains are available in the GenBank database and the accession numbers can be found in Appendix A. Nevertheless, none of the sequences or combinations of sequences could definitively identify the strains, so most ended up with an ambiguous taxonomic name with multiple independent records that shared the same or nearly the same gene sequence similarity. That was why all species were subjected to a phylogenetic analysis.

### 3.2. Sequence Analysis Using GenBank and a Phylogenetic Analysis 

The strains were further analyzed using a phylogenetic approach. Based on the *16S* sequences of the unidentified strains, 33 references strains of mycobacteria and 7 outgroup strains were chosen, and the sequences of the *16S* gene (40 sequences), rpoB gene (32 sequences), and hsp65 gene (26 sequences) of these strains were downloaded to be used in the matrix analysis. See Appendix A for the GenBank access numbers of the sequences from the reference species and outgroup strains. The sequences were aligned in three matrices according to the genes. Three phylogenetic trees were obtained using the phylogeny approach. The *16S* tree demonstrated a consistent topology and identified the *Mycobacterium* species (Appendix A). This tree was selected for species identification. The other two trees (not shown), the *rpoB* and *hsp65* trees, showed highly variable phylogenetic information and were not further used for the purpose of this article.

The monophyletic associations in the *16S rRNA* tree clustered the strains into 16 groups and 5 complexes, with a divergence percentage of 1.328% in the groups (see Appendix A). If the divergence values of each species in relation to the samples were between 0 and 1.328%, they were considered as species under the monophyletic association. If the values slightly exceeded this range, they were considered as NR (near) the corresponding species, and if the associations were not close to any species, they were considered non-identifiable.

Appendix A shows the results obtained for each strain along with the correct species identification of the pathogens. Here, 83.33% (40/48) of the total strains were identified or closely related to a species with this phylogenetic approach. The *16S rRNA* gene analysis only allowed the identification of 68.75% (33/48) of the strains. The tree strains were also identified via rpoB (sample LTP1387 as near *M. senegalense* and *M. farcinogenes* and LTQ2392–LTS855 as *M. conceptionense*). The *hsp65* gene allowed us to identify LTR2603, LTS244, and LTO3295 as *M. fortuitum* and LTN4726 as *M. abscessus*. Remarkably, five “acid-fast” strains were not identified with sequencing as *Mycobacterium* species but were identified as belonging to the genera *Tsukamurella, Nocardia*, and *Gordonia*. See Figure 1 and Figure 2 for examples of the identification using a phylogenetic approach for some of the strains.

## 4. Discussion 

The identification with the sequences of a partial *16S*, *hsp 65*, or *rpoB* gene or with a phylogenetic approach showed a variety of acid-fast species isolated from patients in a clinical laboratory. Most species were previously not properly identified or could not be identified with the PRA technique, a commonly used technique in many clinical laboratories. Additionally, the *16S rRNA* sequencing alone or the sequencing of the *hsp65* and *rpoB* genes and search for a BLAST similarity in GenBank could not assign a species name to most strains. Therefore, the identifications were made with a phylogenetic analysis of the *16S* gene. The phylogenetic analysis improved the identification of the strains by placing them close to a group or complex and provided us with the identification based on their evolutionary relationship.

The identification of our isolates showed that many species are hardly mentioned in the medical literature and are relatively unknown as infectious agents for humans. Examples include the species *M. porcinum, M. simiae, M. thermoresistibile, M. neoaurum,* and *M. szulgai* [22,23]. Some strains (Appendix A and Figure 1 and Figure 2) were unidentifiable because the percentages of divergence were considerably higher than 1.328%, so these could be new mycobacterial species or variants of already known species, or the references sequences were not considered in the phylogenetic analysis. This needs further investigation.

Furthermore, although we expected to identify only mycobacteria because only AFB were used, other species of the genera *Tsukamurella*, *Nocardia*, and *Gordonia* were found. These genera also have mycolic acid in their cell wall, like mycobacterial species [24], which explains their acid-fast properties. In the literature, these bacteria are called “partially or weakly acid-fast”. However, in our laboratory, this “partial acid-fast” nature is hard to differentiate from bacteria with real “acid-fast” characteristics. When viewed under the microscope, short acid-fast red rods were seen by the laboratory technician, who has more than 10 years of experience. Moreover, when we became aware of “weak acid-fast” organisms in our collection of strains, we grew the strains again from the glycerol preparations stored at minus 70 °C. Once again, only acid-fast rods were seen under the microscope. The incorrect identification of the strains in this study could have had grave repercussions for the patients. All isolates came from patients with clinical signs and symptoms that can be confused with tuberculosis. We do not know with which antibiotics these patients were treated with, but incorrect identification can lead to inappropriate antibiotic prescriptions and does not permit species-specific treatment. A correct identification is also important for epidemiological purposes (transmission and infection source studies) and can lead to an underestimation of the importance of the virulence of some species.

The reason we identified these bacteria as acid-fast and consequently as mycobacteria could possibly be found in the composition of our Ziehl–Neelsen stain or decolorizing solution. We used a more concentrated 1% carbol–fuchsine solution (a mixture of phenol and basic fuchsin) in the staining procedures as recommended by the Stop TB Partnership [25], while other ZN staining protocols use a 0.3% solution of carbol–fuchsine [26,27]. It is well established that a higher concentration of fuchsine produces a better staining of the mycobacterial cells [28,29]. Additionally, the phenol concentrations and acid–alcohol solutions vary in the Ziehl–Neelsen staining protocols [30,31,32]. Moreover, the efficiency of the Ziehl–Neelsen method for destaining acid-fast bacteria depends on the composition of the solutions. We used 3% HCL in 95% ethanol as a destaining solution, but 6% HCL has also been recommended or 20–25% sulfuric acid [25,26,27]. The staining and destaining of “partial or weak” AFB with the different solutions has never been evaluated. Despite its application for well over a century, the ZN technique remains non-standardized, and we believe that standardization is urgently required and should incorporate the partially acid-fast genera, which include *Nocardia, Tsukamurella,* and *Gordonia*. These genera are uncommon pathogens and are prone to being misdiagnosed as mycobacteria [33]. An example in our clinical laboratory was strain LTN436, a *Tsukamurella pulmonis* strain, which was isolated from a sputum sample after a decontamination step and grown on L-J medium. The patient was misdiagnosed as infected with a *Mycobacterium* spp. Such an error may be common among lung infections caused by *Tsukamurella* species because the disease manifestation shares a striking similarity with mycobacterial infections [33]. Similarly, strain LTS2331, with a preliminary identification as a *Nocardia* spp., was eventually identified as *Gordonia sputi.* Only in recent years have *Gordonia* species been recognized as emerging pathogens [34]. Globally, from 1996 to 2015, only 16 cases of infections by *Gordonia sputi* were reported, and these were associated with infections from catheters or with lung disease in immunocompromised patients [35]. *Gordonia* spp. are environmental bacteria whose implication in human disease seems to be increasing [36]. Finally, two isolates were respectively identified as *Nocardia puris* and *Nocardia cyriacigeorgica*. The genus *Nocardia* comprises opportunistic pathogens that cause a variety of clinical diseases in immunocompromised people, mostly pulmonary infections [37]. *Nocardia cyriacigeorgica* has been isolated before from lung infections in patients in Venezuela [38]. However, the isolation and identification of *Nocardia puris* as well as the other “weak” acid-fast microorganisms—*Tsukamurella pulmonis* and *Gordonia sputi*—are the first reports of human infection in Latin America. 

## 5. Conclusions

A high variety of mycobacterial strains were identified, with many species rarely described as pathogens in the medical literature. Moreover, acid-fast microorganisms of other genera, also rarely mentioned as pathogens, were found, and three species were identified that have never been reported before in Latin America, namely *Nocardia puris*, *Tsukamurella pulmosis*, and *Gordonia sputi*. We also found that several of the mycobacteria species of this study could not be identified with a “one gene only” sequence mainly due to the great diversity of new species and groups of species and the absence of reference sequences in GenBank, but that the interpretation of the *16S* sequences data with a phylogenetic approach in general permits the identification of genus and species groups based on the closest evolutionary relatives. For the clinical laboratory, a simpler straightforward method for the identification of acid-fast microorganisms is needed. This method could be based on the sequencing of a particular gene or a computational phylogenetic algorithm to identify the acid-fast microorganism. For this purpose, and in development in our lab, is an online web-based identification system that will use a phylogenetic analysis (including all AFB genera) based on *16S rDNA* sequences, phylogenetic trees, and close relatives. This database will be constantly updated with genetic information concerning the mycobacterial species and will permit the identification based on the relationships with its closest relatives. Several online databases that track the pathogen identification and evolution processes in real time are already available, including the platform Nextstrain (available online: https://nextstrain.org/; accessed on 20 September 2022),which tracks the pathogenic evolution of several viruses and *Yersinia pestis,* or the Pango Network (available online: https://www.pango.network/; accessed on 20 September 2022, developed for identifying SARS-CoV-2 genetic lineages of epidemiological relevance. 

## 6. Limitations of This Study

We used no clinical data and did not perform follow-up checks on the patients. Most isolates were from many years ago and telephone contact was lost. Thus, we were unable to determine what effect the incorrect identification of these AFB had on the disease outcomes. 

## Figures and Tables

**Figure 1 pathogens-11-01159-f001:**
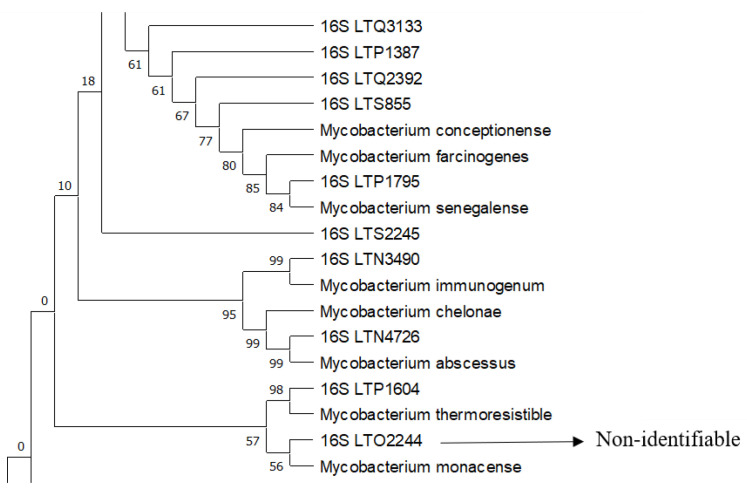
A segment of the *16S rRNA* tree showing complex 2 and the identification of the laboratory strains indicated with *16S* using a phylogenetic approach with sequences of reference strains. Phylogenetic construction was performed via ML and T92+G+I using an MP as a guide tree, with the robustness of the bootstrapping set at 500 pseudo-replicates.

**Figure 2 pathogens-11-01159-f002:**
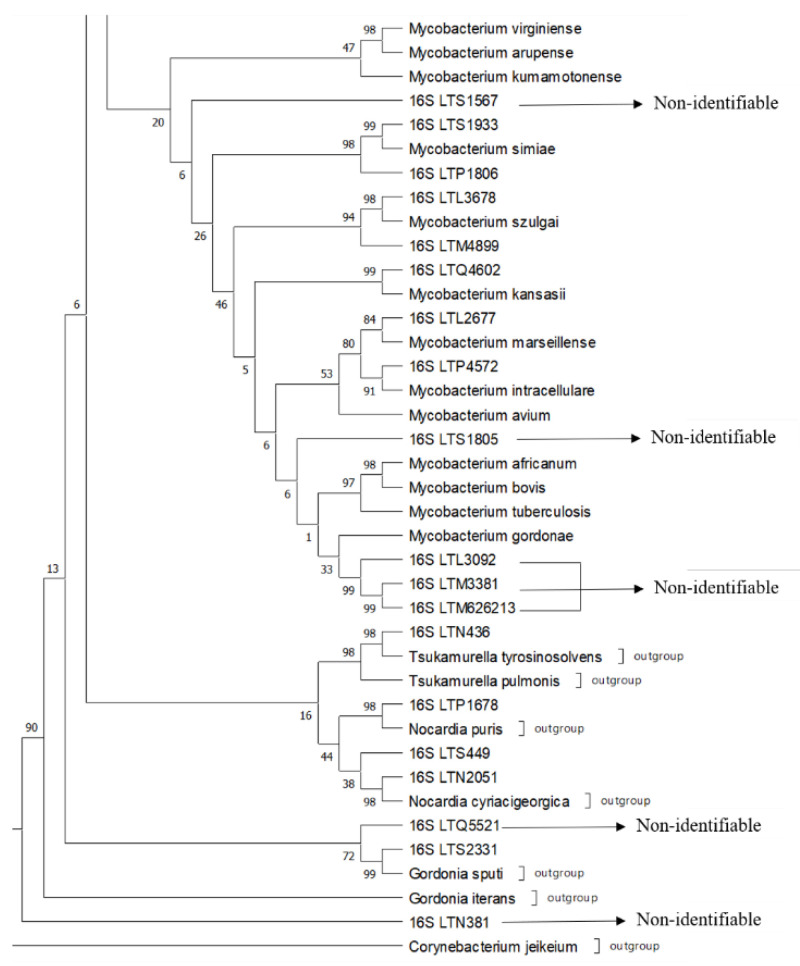
A segment of the *16S rRNA* tree showing complexes 4 and 5, outgroups of the laboratory strains indicated for identification with *16S* using a phylogenetic approach with sequences of reference strains and outgroup strains. The phylogenetic construction was performed using ML and T92+G+I using an MP as a guide tree, with the robustness of the bootstrapping set at 500 pseudo-replicates.

**Table 1 pathogens-11-01159-t001:** Oligonucleotide primers used for the sequencing of parts of the *16S*, *rpoB*, and *hsp65* genes of the mycobacteria in this study. The primers used were previously described [17,18,19].

Gene	Primer	Annealing Temperature	Sequence	
** *16s* **	A1F	63 °C	5′-ctggctcaggacgaacgctg-3′	*Forward*
	54R		5′-tctagtctgcccgtatcgccc-3′	*Reverse*
** *rpob* **	RPO5’	65 °C	5′-tcaaggagaagcgctacga-3′	*Forward*
	RPO3’		5′-ggatgttgatcagggtctgc-3′	*Reverse*
** *hsp65* **	Tb11	65 °C	5′-accaacgatggtgtgtccat-3′	*Forward*
	Tb12		5′-cttgtcgaaccgcataccct-3′	*Reverse*

## Data Availability

All sequence data are published in GenBank.

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
