# Peer review of "The Re-Identification of Previously Unidentifiable Clinical Non-Tuberculous Mycobacterial Isolates Shows Great Species Diversity and the Presence of Other Acid-Fast Genera"

_pathogens, 2022, doi:10.3390/pathogens11101159_

Round 1

Reviewer 1 Report

Thank you for giving me the opportunity to review this manuscript.

The information presented by the manuscript was clear, but its innovation could be improved and should provide more new insights. I have following comments.

 Major: The clinical significance of the NTM re-identified was not clear. Whether these pathogens caused disease? What are the symptoms?

Minor:

Introduction

(1) The text “NTM are facultative intracellular pathogens that are not transmitted from person-to-person but acquired from the environment” in the first paragraph is too absolute. It has been reported that person-to-person transmission of M. abscessus between patients with cystic fibrosis. (Bryant J M, Grogono D M, Rodriguez-Rincon D, et al. Emergence and spread of a human-transmissible multidrug-resistant nontuberculous mycobacterium[J]. Science, 2016,354(6313):751-757.)

The text “which has led to the identification of over 170 recognized species of Mycobacterium” in the first paragraph. To our knowledge, more than 200 NTM strains have been identified.

(2) Streamline the content and keep it to about 400 words. Emphasis on " identification ", not "epidemiology".

(3) The third paragraph can be used to summarize the purpose of the study in a few sentences. The remainder can be deleted because it is repetitive with the “Conclusion”.

(4) Abbreviations that first appear in the article need to be expanded.

2. Result

The description of the method of constructing the phylogenetic tree is needed in the Figure.

3. Discussion

(2) The findings of the study and its significance should be highlighted.

(3) The references should be updated.

(4) Please add the limitations of the study.

4. Conclusions

(1) The first paragraph can be placed in the “Discussion”.

(2) The text “Especially in these cases, it is important to mention that incorrect identification can lead to inappropriate antibiotic prescriptions. A correct identification permits physicians to initiate a prompt and species-specific treatment and is also important for epidemiological purposes (transmission and infection source studies)” is similar to the contents of the “Introduction” and can be deleted.

Reviewer 2 Report

Bacteria stained with EZN and NTM and also their epidemiology is not really clear. It is rare for them to be documented as a etiological agent because they are very difficult to identify. This is valuable work.

Abbreviation of restriction fragment length polymorphism is known as RFLP generally. Is there any specific reason to change to??

Discussion and Material and methods sections were reversed?? There was an error in the order of the sections.

Discussion is not sufficient. I suggest that it be a little more detailed such as discussing those isolated from environmental and human samples.
